# HLA and Non-HLA Factors for Donor Selection in Hematopoietic Stem Cell Transplantation with Post-Transplant Cyclophosphamide GvHD Prophylaxis

**DOI:** 10.3390/cells13242067

**Published:** 2024-12-14

**Authors:** Hiroko Shike, Aiwen Zhang

**Affiliations:** 1Department of Pathology, Penn State Milton S. Hershey Medical Center, Hershey, PA 17033, USA; 2Cleveland Clinic, Allogen, Pathology & Laboratory Medicine Institute, Cleveland, OH 44195, USA; zhanga2@ccf.org

**Keywords:** PTCy, HLA, DSA, donor age, haploidentical, unrelated

## Abstract

Human leukocyte antigen (HLA) mismatches in stem cell transplantation can be well-tolerated with the use of post-transplant cyclophosphamide (PTCy) for graft-versus-host-disease (GvHD) prophylaxis. Haploidentical (Haplo) and HLA-mismatched unrelated donors become acceptable donors. This review focuses on Haplo and unrelated donor selection in the context of PTCy-transplant for hematological malignancy, in comparison with conventional GvHD prophylaxis. Evaluating patient’s donor-specific antibody (DSA) is critical in donor selection regardless of donor type or the use of PTCy. High DSA levels and positive C1q increase the risk of engraftment failure and unsuccessful desensitization. On the other hand, the degree of donor HLA matching is less critical under PTCy compared to conventional GvHD prophylaxis. Donor age was found to be important, as younger donors improve survival outcomes. HLA-B leader match appears to be preferable. The impacts of donor gender, donor cytomegalovirus serostatus, and ABO mismatch are unclear or non-significant. Additionally, available studies suggest that, in PTCy-transplant, preferred Haplo-donors are HLA class II mismatched (*DRB1* mismatch and *DPB1* non-permissive), siblings or offspring over parents, and if parent, father over mother, while preferred unrelated donors are HLA class I matched. Further study is warranted.

## 1. Introduction

Human leukocyte antigen (HLA) disparity was the major immunological barrier in allogeneic stem cell transplantation (SCT) under cyclosporin or tacrolimus-based conventional graft-versus-host-disease (GvHD) prophylaxis. The importance of HLA high-resolution matching at *HLA-A*, *B*, *C*, and *DRB1* (match count 8/8) was well-documented from (1) comparable survival outcomes from HLA-matched sibling donors (MSDs) and matched unrelated donors (MUDs, 8/8 match) [1,2,3], and (2) inferior outcomes from mismatched unrelated donors (MMUDs, <8/8 match) compared to MUDs [4,5,6]. Thus, in unrelated donor transplantation, HLA-matched donors were recommended by the National Marrow Donor Program (NMDP) and the Center for International Blood and Marrow Transplant Research (CIBMTR) in 2019 [7]. For many patients without available MSDs or MUDs, the options were MMUDs, haploidentical (Haplo) donors, or cord blood units, the donors considered as alternatives due to inferior survival outcomes.

HLA disparity from alternative donors was well-tolerated with the use of new GvHD prophylaxis, post-transplant cyclophosphamide (PTCy) [8], or anti-thymocyte globulin (ATG) [9]. Consideration of HLA matching is no longer included by the Haplo-donor selection consensus recommendations, the latest published for PTCy-transplant by the European Society for Blood and Marrow Transplantation (EBMT) in 2020 [10] and for ATG transplant by the Chinese Society of Hematology (CSH) in 2021 for ATG transplant [11]. Additionally, the use of PTCy or ATG was shown to improve unrelated donor transplant outcomes in prospective multicenter trials [12,13]; thus, it was adopted as the standard GvHD prophylaxis by the consensus recommendation by the EBMT in 2024 [14]. Recent studies suggest that HLA mismatch may affect differently depending on GvHD prophylaxis regimens (PTCy or ATG) and donor relations (Haplo, MSD, or unrelated) [10,11]. However, the data are insufficient to compare the impact of HLA and non-HLA factors and to rank them for hierarchy in donor selection. This review assesses the factors influencing Haplo and unrelated donor selection for PTCy-transplant in hematological malignancies based on recent findings and published consensus recommendations.

## 2. Donor Types by Relation and HLA-Match

MSDs, MUDs, MMUDs, and Haplo are all acceptable donors in PTCy-transplant. 

Transplant from HLA mismatch donors (MMUDs/Haplo) were shown to have comparable survival outcomes compared to the transplant from HLA match donors (MSDs/MUDs) [15,16,17,18,19] or HLA-mismatched donors (MMUDs vs. Haplo) [20] when both groups received PTCy. However, some reported inferior survival with Haplo compared to MUDs [21,22] as well as to MSDs [23]. While the hierarchy among these donor types is unclear, even with the use of PTCy, HLA-matched donors (MSDs and MUDs) may still be preferred if available.

## 3. HLA Factors

### 3.1. Donor-Specific Antibody (DSA)

Antibody-mediated rejection is one of the primary mechanisms contributing to graft failure regardless of donor type or GvHD prophylaxis regimen [24,25,26]. Since the risk of primary graft failure increases when a patient has DSA, an antibody to the mismatched donor’s HLA antigens, consensus guidelines generically recommend avoidance of DSA for unrelated/cord donor transplant with conventional GvHD prophylaxis (NMDP/CIBMTR 2019) [7], Haplo-PTCy-transplant (EBMT 2020) [10], and Haplo-ATG transplant (CSH 2021) [11]. The risk of DSA-mediated engraftment failure is especially higher in child-to-mother transplants since patients might have developed DSA from pregnancies decades ago [27].

The detailed studies on DSA indicated that primary graft failure was associated with DSA > 5000 mean fluorescence intensity (MFI) tested by the commonly used and commercially available Luminex-based solid-phase immunoassay [28,29]. Patients with DSA < 5000 MFI experienced no adverse outcomes without desensitization [30]. The risk of graft failure increased significantly with DSA > 10,000 MFI [29] or with complement-fixing [28] detected by the Luminex C1q assay. Successful engraftment was reported when post-desensitization DSA reduced to the levels that are negative by flow cytometry crossmatch [31] or <5000 MFI [32]. Responses to desensitization were reported for DSA levels <10,000 MFI before treatment with rituximab [33] or <20,000 MFI before treatment with plasma exchange, rituximab, intravenous immunoglobulin, and a donor buffy coat [34]. However, the use of MFI must be taken with caution [35] because (1) antigen saturation of the assay due to excess HLA antibody can cause the underestimation of MFI; (2) denatured antigens or treatment interference (for example, intravenous immunoglobulin) can cause false positive [36]; and (3) “interfering factors” competing with HLA antibody can cause falsely low MFI. To overcome the saturation issue, a dilution or titration study can improve the evaluation of DSA levels [35]. To resolve denatured antigen issues, antibody results by alternative methods should be correlated. To avoid interfering factors, serum treatment such as ethylenediaminetetraacetic acid is recommended [37].

### 3.2. HLA Match Degree

In unrelated donor transplants under conventional GvHD prophylaxis, HLA mismatches at *HLA-A*, *B*, *C*, and *DRB1* decreased survival without significant difference between low-resolution mismatch (antigen level) and high-resolution mismatch (P-group level or above [38]) [4]. Thus, also in Haplo-PTCy and unrelated donor-PTCy-transplant, HLA match degree, a number of matched HLA loci was evaluated at the four loci without differentiating by resolutions. In Haplo-PTCy-transplant, HLA match degree was reported with no significant impacts on GvHD or survival [39,40,41,42,43]. Therefore, the consensus recommendation (EBMT 2020) for Haplo-PTCy-transplant includes no criteria for HLA match degree [10]. Meanwhile, the best cumulative survival was reported with three class II mismatches (*HLA-DRB1*, *DQB1*, and *DPB1* non-permissive) [44,45].

In unrelated donor PTCy-transplant, a prospective clinical trial by NMDP/CIBMTR reported comparable survival between 7/8 match and 4/8–6/8 match using bone marrow (BM) grafts [46] and peripheral blood (PB) grafts [47]. An EBMT working party also reported comparable survival between 8/8 match and 7/8 match using PB grafts [48].

### 3.3. HLA-Locus-Specific Mismatch

In Haplo-PTCy-transplant, associations were reported as follows: (1) *HLA-A* mismatch with lower acute GvHD risk [49] or higher chronic GvHD risk [44], (2) *HLA-C* mismatch with higher chronic GvHD risk [41], (3) *HLA-DRB1* mismatch (over match) with improved survival [44,45], lowered relapse risk [41], no impact [39], or grade ≥2 acute GvHD risk (if antigenic mismatch) [43], and (4) *DPB1* non-permissive (over permissive) with improved survival [41,44,45]. Thus, in Haplo-PTCy-transplant, HLA class II mismatch may be preferred. On the other hand, another EBMT working party reported inferior survival associated with HLA class I mismatch (but not class II mismatch) in unrelated PTCy-transplant [50]. Thus, in unrelated donor PTCy-transplant, HLA match degree is not critical, but HLA class I match may be preferred. Further study is warranted.

### 3.4. HLA-B-Leader

The HLA-B-leader is dimorphic, methionine (M) or threonine (T) at the amino acid position −21, encoded by the *HLA-B* exon 1. Although this is not a structural moiety of the mature HLA-B molecule, the HLA-B-leader (M) binds HLA-E and controls the natural killer cell response via the inhibitory NKG2A receptor [51]. B-leader match (the term for the same B-leader in the mismatched *HLA-B*) was reported to be associated with lower acute GvHD risk in HLA-B-mismatched unrelated donor transplantation under conventional GvHD prophylaxis [52]. Similarly, B-leader match was reported to improve survival in Haplo-PTCy-transplantation [41,49]. The HLA-B leader assessment tool, for research purposes only (BLEAT, https://bleader.nmdp.org/ (accessed on 1 December 2024)) assists in the prediction of B-leader genotyping and match status [52,53].

Among the four combinations of Haplo-PTCy groups by B-leader and *DRB1* (mis)match status, the best survival in the B-leader-matched/*DRB1*-mismatched group and the poorest survival in the B-leader-mismatched/*DRB1*-matched group [54] were reported, suggesting combined impacts. To predict the probability of the disease-free survival among multiple Haplo-donors based on *DRB1*/*DQB1* [41,55], *DPB1* [56], and B-leader [41,52] matching status, a research web tool is provided (https://haplodonorselector.b12x.org/v1.0/ (accessed on 1 December 2024)) by the CIBMTR [41]. Further study is warranted for the application in clinical donor selection.

Donor B-leader genotypes (M or T) are not shown to be associated with transplant outcome, although patient’s B-leader (M) was associated with superior survival and lower relapse in Haplo-PTCy-transplant for lymphoid malignancy (not for myeloid) [57]. Similarly, in unrelated donor transplantation with conventional GvHD, patient’s B-leader (M), but not donor’s B-leader (M/T) was associated with higher non-relapse mortality in the presence of *DQB1* mismatch and grade III-IV GvHD risk in the presence of *DRB1* mismatch [58]. Thus, in PTCy-transplant, donor B-leader genotyping is not considered in donor selection.

## 4. Non-HLA Factors

### 4.1. Donor Age

The importance of unrelated donor age was well-documented under conventional GvHD [6,59]. Thus, the unrelated donor selection guideline with conventional GvHD prophylaxis (NMDP/CIBMTR 2019) prioritizes younger donors over older donors [7]. In unrelated donor PTCy-transplant, increasing donor age was also reported as an independent risk factor, which is more significant than HLA mismatch [48].

The Haplo-PTCy consensus recommendation (EBMT 2020) [10] also prioritizes younger donors over older donors because increasing donor age was associated with inferior survival [19,60,61,62], higher acute but not chronic GvHD [60], and higher non-relapse mortality with lower relapse risk [39,63]. However, in Haplo-PTCy transplant, an assessment of outcomes based on donor age is challenging, as it is confounded by donor–recipient relationship/kinship and patient age [45,60,64]. When the influences from the two factors, patient age and parent donor were removed, increasing donor age was still a significant risk factor in one study [60] but not in others [45,64]. Additionally, increasing donor age was reported as a risk factor of poor mobilization in allogenic peripheral blood stem cell transplantation [65,66,67]. Overall, younger donors should be prioritized if available.

### 4.2. Haplo-Donors by Relationship

The Haplo-PTCy consensus recommendation (EBMT 2020) prioritizes offspring and siblings over parent donors [10]. Most reported inferior outcomes with parent donors in PTCy-transplant, including higher graft failure [64], inferior survival [45,60], and higher relapse risk (only in patients ≤40 years but not >40 years) [63]. One study reported no impacts on the donor–recipient relationship after controlling for donor age and patient age [39]. Graft failure and survival did not differ between sibling and offspring donors [64]. Thus, offspring and siblings are equally preferred over parents. In addition to first-degree Haplo-donors, non-first-degree Haplo-donors (niece, nephew, cousin, etc.) were also reported as acceptable with comparable outcomes to first-degree Haplo-donors [68].

The Haplo consensus (EBMT) [10] did not recommend one over the other due to insufficient data regarding outcome differences between donor choice between mother or father. In ATG-transplant, the recommended choice is father over mother in hematological malignancy (CSH 2021) [11]. In Haplo-PTCy transplant, many studies have reported that mother-to-child transplant was associated with inferior survival [69,70], increased GvHD risk [70], and increased relapse risk [69]; meanwhile, the other research results have shown no significant impacts on survival [39,60], or decreased relapse risk [71]. Father may be used over mother if available, although there are inconsistent conclusions in the literature.

A Haplo-sibling and a patient share either a paternal or a maternal HLA haplotype. Donor has a non-inherited maternal antigen (NIMA)-mismatch if the shared haplotype is paternal or a non-inherited paternal antigen (NIPA)-mismatch if the shared haplotype is maternal. It has been hypothesized that a sibling donor with a NIMA-mismatch is immunogenically more tolerated due to exposure to maternal antigens in utero [72]. In ATG-transplant, NIMA-mismatch over NIPA-mismatch is recommended (CSH 2021) [11] for rapid regulatory T-cell recovery [73], a lower risk of acute GvHD [73,74], and better survival [74]. On the other hand, no recommendation has been made for donor NIMA/NIPA-mismatch status in PTCy-transplant (EBMT 2020) [10] due to insufficient data. Recent studies reported less acute GvHD [49], suggesting a potential modest benefit of NIMA-mismatch in PTCy-transplant.

### 4.3. Donor Gender

In transplants from a female donor to a male patient (female-to-male), increased GvHD risk was hypothesized because female donor T lymphocytes may recognize the patient’s minor histocompatibility antigens encoded on the Y chromosome. Donor gender consideration is not included in the recommendation for unrelated donor SCT guidelines under conventional GvHD prophylaxis (NMDP/CIBMTR 2019) [7] based on the lack of significant survival impacts in two large independent cohorts [6]. On the other hand, male-to-male over female-to-male is recommended in Haplo-PTCy transplant (EBMT 2020) [10] based on inferior survival in 48 female-to-male transplants compared to 137 other transplants [40] and a higher ≥2-grade acute GvHD in transplant from female donors [43]. However, recent studies reported inconsistent findings. Male-to-male transplant was associated with inferior survival in AML patients [75]. Female-to-male transplant was associated with comparable survival [41,42,60], a higher chronic GvHD risk [41,63], lower relapse risk [71], or comparable GvHD risk [42] in Haplo-PTCy transplant, while comparable outcomes were reported in unrelated donor PTCy transplant [48]. Further studies are warranted.

### 4.4. Cytomegalovirus (CMV) Serostatus

Donor CMV serostatus was not included in the recommendation for unrelated donor SCT guidelines under conventional GvHD prophylaxis (NMDP/CIBMTR 2019) [7] due to its insignificant impact on survival in a large study that specially addressed donor characteristics [6]. Donor CMV serostatus was also not included in the Haplo-PTCy consensus recommendation (EBMT 2020) [10] due to conflicting results. Many publications reported that inferior survival was associated with patient’s [64,76] but not donor’s CMV seropositivity [64,76,77], except for the reports of inferior survival using seronegative donors for seropositive recipients [45,75]. Regardless of donor or recipient serotypes, PTCy use was shown to increase CMV infection in the transplant from Haplo, MSDs [23], and MUDs [78]. Aggressive prevention was recommended in all receiving PTCy [23,78] since CMV infection increases the risk of chronic GvHD, negating the protection by PTCy [23].

### 4.5. Blood Type ABO Compatibility

ABO compatibility was not included in the recommendations for unrelated donor SCT guidelines under conventional GvHD prophylaxis (NMDP/CIBMTR 2019) [7]. This was because ABO mismatch was not detected to have a significant impact on survival in a large study that specially addressed donor characteristics [6], although others reported poor survival for minor/bidirectional mismatch (from unrelated but not related donors) [79], minor/major mismatch [80], or major mismatch [81]. The Haplo-PTCy consensus recommendation (EBMT 2020) [10] discussed that the impact of ABO mismatch on transplant outcomes appears to be variable in different transplant settings. ABO mismatch was evaluated but not detected as a risk factor in the Haplo-PTCy cohorts, regardless the use of mostly BM grafts [64] or PB grafts [39,45]. The impact of ABO mismatch in unrelated-PTCy SCT is unknown. At this time, there is no evidence to require ABO compatibility in PTCy transplant.

## 5. Conclusions

Matched donors are preferred if available. However, Haplo and MMUDs are also acceptable because HLA matching is less critical under PTCy. The impacts of HLA and non-HLA factors in Haplo-PTCy-transplant, unrelated-PTCy-transplant, and unrelated donor transplant under conventional GVHD prophylaxis are summarized (Table 1). The evaluation of DSA is important. In PTCy-transplant, preferred donors should be with younger age and matched HLA-B leader. Additionally, in Haplo-PTCy transplant, HLA class II mismatch (*DRB1* mismatch and *DPB1* non-permissive), a sibling or an offspring over a parent, and if a parent is chosen, a father over a mother may be preferred. In unrelated donor PTCy-transplant, HLA class I match may be preferred. The impacts of other HLA and non-HLA factors appear to be modest, insignificant, or unclear. Further studies are warranted.

## Figures and Tables

**Table 1 cells-13-02067-t001:** Impact of HLA and non-HLA factors.

GvHD Prophylaxis	PTCy		Conventional
Donor Types	Haploidentical	Unrelated	Unrelated
**Latest consensus recommendation or guideline**	EBMT 2020 [10]	Not published	NMDP/CIBMTR 2019 [7]
**HLA factors**			
**Donor-specific antibody**	Risk [26]	Risk	Risk [24]
**HLA mismatch degree**	No impact [39,40,41,42,43]	No impact [46,47,48]	Risk (HLA-A, B, C, DRB1) [4,6]
**Class I mismatch**	Potential risk HLA-A [44,49]; HLA-C [41]	Potential risk [50]	
**Class II mismatch**	Potential benefit [41,44,45] No impact [39] Potential risk [43]	No impact [50]	
**B-leader match**	Potential benefit [41,49]	Insufficient data	Potential benefit [52]
**Non-HLA factors**			
**Older donor**	Risk [19,60,61,62,65,66,67]	Risk [65,66,67]	Risk [6,65,66,67]
**Offspring/sibling vs. parent**	Offspring/sibling is better [45,60,63,64]	Not applicable	Not applicable
**Father vs. mother**	Father better [69,70] No difference [39,60]	Not applicable	Not applicable
**NIMA-M vs. NIPA-M**	Potential benefit of NIMA-M [49]	Not applicable	Not applicable
**Female-to-Male**	Potential risk [40,41,43,63] No impact [41,42,60]	No impact [48]	No impact [6]
**Donor CMV serostatus**	No impact [64,76,77]	Insufficient data	No impact [6]
**ABO match**	No impact [39,45,64]	Insufficient data	No impact [6]

**Abbreviation**: CMV: cytomegalovirus, EBMT: European Society for Blood and Marrow Transplantation; GvHD: graft versus host disease; NIMA-M: non-inherited maternal antigen-mismatch; NIPA-M: non-inherited paternal antigen-mismatch; NMDP/CIBMTR: National Marrow Donor Program/Center for International Blood and Marrow Transplant Research; PTCy: post-transplant cyclophosphamide.

## Data Availability

No new data were created or analyzed in this study.

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
