# Peer review of "HLA and Non-HLA Factors for Donor Selection in Hematopoietic Stem Cell Transplantation with Post-Transplant Cyclophosphamide GvHD Prophylaxis"

_cells, 2024, doi:10.3390/cells13242067_

Round 1

Reviewer 1 Report

Comments and Suggestions for Authors

The paper is well-written and covers all the elements involved in donor selection. 

I suggest to add the concept that MFI might vary according to the vendor, and this is a limitation in the inter-laboratory interpretation. 

Concerning donor age, it is important to select a younger one also because of better stem cell mobilization and fewer complication during apheresis: this is an important point, to be addressed.

Authors never mention if mismatches are at one-field (antigenic) or two-field (allelic), therefore I suggest to include  this information.

Finally, a table should be provided, with the factors (HLA and non-HLA) and the message together with the respective literature. 

Author Response

I suggest to add the concept that MFI might vary according to the vendor, and this is a limitation in the inter-laboratory interpretation.

All MFI values cited in this review (Ref: 27-35) as well as in most publication use LABScreen Single Antigen from ThermoFisher (previously One Lambda, CA).  Vendor variation is not a major issue.  

To clarify this point, we inserted (green) to page 5, line 24; “….DSA >5000 mean fluorescence intensity (MFI) using tested by a commonly-used and commercially-available Luminex-based solid-phase immunoassays [28, 29].

Major limitations of MFI are as described (page 6, line 1 in the Revised version with Track Change); 1) antigen saturation of assay due to excess HLA antibody can cause underestimation of MFI, 2) denatured antigens or treatment interference (for example, intravenous immunoglobulin) can cause false positive[36], 3)  “interfering factors” competing with HLA antibody can cause falsely low MFI. 

2. Concerning donor age, it is important to select a younger one also because of better stem cell mobilization and fewer complication during apheresis: this is an important point, to be addressed.

Following sentence was inserted to page 8, line 26: “Additionally, increasing donor age was reported as a risk of poor mobilization in allogenic peripheral blood stem cell transplantation [65],[66],[67].”

Additionally, we modified (page 8, line 28) from “younger donor may still be prioritized…” to  “younger donor should be prioritized…”.

Added References

[65] Vasu S, Leitman SF, Tisdale JF, Hsieh MM, Childs RW, Barrett AJ et al. Donor demographic and laboratory predictors of allogeneic peripheral blood stem cell mobilization in an ethnically diverse population. Blood 2008; 112(5): 2092-2100. e-pub ahead of print 20080603; doi: 10.1182/blood-2008-03-143677

[66] Teipel R, Schetelig J, Kramer M, Schmidt H, Schmidt AH, Thiede C et al. Prediction of hematopoietic stem cell yield after mobilization with granulocyte-colony-stimulating factor in healthy unrelated donors. Transfusion 2015; 55(12): 2855-2863. e-pub ahead of print 20150716; doi: 10.1111/trf.13239

[67] Piccirillo N, Putzulu R, Metafuni E, Massini G, Fatone F, Corbingi A et al. Peripheral Blood Allogeneic Stem Cell Mobilization: Can We Predict a Suboptimal Mobilization? Transfus Med Rev 2023; 37(2): 150725. e-pub ahead of print 20230517; doi: 10.1016/j.tmrv.2023.150725

3. Authors never mention if mismatches are at one-field (antigenic) or two-field (allelic), therefore I suggest to include this information.

By definition, two-field is high resolution, and  four-field is allelic resolution.  Following sentence was inserted to page 6, line 11.

In unrelated donor transplant under conventional GvHD prophylaxis, HLA mismatches at HLA-A, B, C, and DRB1 decreased survival but without no significant difference between low-resolution mismatch (antigen level) and high-resolution mismatch (P-group level or above [38]) [4]. Thus, also in Haplo-PTCy and unrelated donor-PTCy transplant, HLA match degree, a number of matched HLA loci was evaluated at the four loci without differentiating by resolutions. “

To be consistent for the four loci evaluated (HLA-A, B, C, and DRB1), we modified the following sentence (page 6, line 27), “An EBMT working party also reported comparable survival between 10/10 match vs. 9/10 match using PB graft [48]” to “An EBMT working party also reported comparable survival between 8/8 match vs. 7/8 match using PB graft [48]”.  

Added References

[4] Lee SJ, Klein J, Haagenson M, Baxter-Lowe LA, Confer DL, Eapen M et al. High-resolution donor-recipient HLA matching contributes to the success of unrelated donor marrow transplantation. Blood 2007; 110(13): 4576-4583. e-pub ahead of print 2007/09/06; doi: blood-2007-06-097386 [pii]

[37] Nunes E, Heslop H, Fernandez-Vina M, Taves C, Wagenknecht DR, Eisenbrey AB et al. Definitions of histocompatibility typing terms. Blood 2011; 118(23): e180-183. e-pub ahead of print 2011/10/18; doi: blood-2011-05-353490 [pii] 10.1182/blood-2011-05-353490

[48] Sanz J, Labopin M, Choi G, Kulagin A, Peccatori J, Vydra J et al. Younger unrelated donors may be preferable over HLA match in the PTCy era: a study from the ALWP of the EBMT. Blood 2024; 143(24): 2534-2543. doi: 10.1182/blood.2023023697

4. Finally, a table should be provided, with the factors (HLA and non-HLA) and the message together with the respective literature.

A table (Table 1) and a following sentence were inserted in conclusion (page 11, line 12). “Impacts of HLA and non-HLA factors in Haplo-PTCy transplant, Unrelated-PTCy transplant, and Unrelated donor transplant under conventional GVHD prophylaxis were summarized (Table 1).” 

Reviewer 2 Report

Comments and Suggestions for Authors

Comments

In the review “HLA and non-HLA factors for donor selection in hematopoietic stem cell transplantation with post-transplant cyclophosphamide GvHD prophylaxis Hiroko Shike et al. present evidence for the significance of HLA donor selection in GvHD prophylaxis using post-transplant cyclophsphamide (PTCy).

The data presented are not conclusive, as already suggested by the authors. Further studies are required with similar cohort sizes and samples of different descent to justify the significance of these factors in the selection of donors for HSCT. However, the review is coherent and adequately covers the current status that characterizes the area of transplantations with GvHD prophylaxis. Just few minor comments suggested:  

·       Rewrite lines 56-58 (grammar issue)

·       Rewrite lines 79-80 (grammar issue)

·       Please mention the genes with the mismatches when bone marrow and peripheral blood was used in unrelated donor PTCy-transplant (lines 116-119)

Author Response

1. Rewrite lines 56-58 (grammar issue): Word grammar check passed. No issue found. 

2. Rewrite lines 79-80 (grammar issue): Word grammar check passed. No issue found. 

3. Please mention the genes with the mismatches when bone marrow and peripheral blood was used in unrelated donor PTCy-transplant (lines 116-119)

To explain the match loci for (page 6, line 25 in the revised version with Track Change) “In unrelated donor PTCy-transplant, …… 7/8 match vs. 4/8-6/8 match using bone marrow (BM) graft [44] and peripheral blood (PB) graft [45]. An EBMT ….. between 10/10 match vs. 9/10 match [46] “, following sentences were inserted (page 6, line 11) “In unrelated donor transplant under conventional GvHD prophylaxis, HLA mismatches at HLA-A, B, C, and DRB1 decreased survival but without no significant difference between low-resolution mismatch (antigen level) and high-resolution mismatch (P-group level or above [38]) [4]. Thus, also in Haplo-PTCy and unrelated donor-PTCy transplant, HLA match degree, a number of matched HLA loci was evaluated at the four loci without differentiating by resolutions. “

To be consistent for the four loci evaluated (HLA-A, B, C, and DRB1), we modified the following sentence (page 6, line 27), “An EBMT working party also reported comparable survival between 10/10 match vs. 9/10 match using PB graft [48]” to “An EBMT working party also reported comparable survival between 8/8 match vs. 7/8 match using PB graft [48]”.   

Added References

[4] Lee SJ, Klein J, Haagenson M, Baxter-Lowe LA, Confer DL, Eapen M et al. High-resolution donor-recipient HLA matching contributes to the success of unrelated donor marrow transplantation. Blood 2007; 110(13): 4576-4583. e-pub ahead of print 2007/09/06; doi: blood-2007-06-097386 [pii]

[37] Nunes E, Heslop H, Fernandez-Vina M, Taves C, Wagenknecht DR, Eisenbrey AB et al. Definitions of histocompatibility typing terms. Blood 2011; 118(23): e180-183. e-pub ahead of print 2011/10/18; doi: blood-2011-05-353490 [pii] 10.1182/blood-2011-05-353490

[48] Sanz J, Labopin M, Choi G, Kulagin A, Peccatori J, Vydra J et al. Younger unrelated donors may be preferable over HLA match in the PTCy era: a study from the ALWP of the EBMT. Blood 2024; 143(24): 2534-2543. doi: 10.1182/blood.2023023697